 

# Excitatory and inhibitory synapse reorganization immediately after critical sensory experience in a vocal learner

Ziqiang Huang[1,2], Houda G Khaled[3], Moritz Kirschmann[1,2,4], Sharon MH Gobes[3], Richard HR Hahnloser[1,2]*

[1]Institute of Neuroinformatics, University of Zurich and ETH Zurich, Zurich, Switzerland; [2]Neuroscience Center Zurich, Zurich, Switzerland; [3]Neuroscience Program, Wellesley College, Wellesley, United States; [4]Center for Microscopy and Image Analysis, University of Zurich, Zurich, Switzerland

**Abstract** Excitatory and inhibitory synapses are the brain's most abundant synapse types. However, little is known about their formation during critical periods of motor skill learning, when sensory experience defines a motor target that animals strive to imitate. In songbirds, we find that exposure to tutor song leads to elimination of excitatory synapses in HVC (used here as a proper name), a key song generating brain area. A similar pruning is associated with song maturation, because juvenile birds have fewer excitatory synapses, the better their song imitations. In contrast, tutoring is associated with rapid insertion of inhibitory synapses, but the tutoring-induced structural imbalance between excitation and inhibition is eliminated during subsequent song maturation. Our work suggests that sensory exposure triggers the developmental onset of goal-specific motor circuits by increasing the relative strength of inhibition and it suggests a synapse-elimination model of song memorization.

DOI: https://doi.org/10.7554/eLife.37571.001

*For correspondence:
rich@ini.ethz.ch

**Competing interests:** The authors declare that no competing interests exist.

## Introduction

During critical period learning early in life, brain structure is regulated not only by intrinsic factors such as age, but also by extrinsic factors including sensory experience (*Andersen, 2003*; *Hubel and Wiesel, 1970*; *Kirkwood et al., 1995*). During this period, neuronal connections are highly susceptible to experience-dependent modifications (*Hensch, 2004*), often manifest as competitive synapse elimination (*Lichtman and Colman, 2000*).

Both excitatory and inhibitory transmissions have important roles in shaping the structural and functional outcomes of sensory experience (*Bear et al., 1990*; *Hata and Stryker, 1994*). However, the requirements on excitatory and inhibitory synapses during critical period learning have remained elusive. Excitatory and inhibitory neurotransmitter systems are involved in severe mental disorders (*Lee et al., 2015*; *Rubenstein and Merzenich, 2003*; *Yizhar et al., 2011*), which is why a better understanding of these systems during developmental learning may be of therapeutic relevance.

We study the influence of sensory experience and of subsequent motor maturation on the structural balance between excitation and inhibition. We use the songbird as a model system because of its highly specialized song system. Similar to human speech development (*Neville and Bavelier, 2002*), the normal development of birdsong requires early sensory experience (*Barrington, 1773*; *Beecher and Brenowitz, 2005*).

Adult birdsong is generated and temporally controlled by interactions among inhibitory and excitatory neurons in HVC (*Aronov et al., 2008*; *Kosche et al., 2015*; *Long and Fee, 2008*; *Mooney and Prather, 2005*; *Nottebohm et al., 1976*). An important role of excitation is to

**eLife digest** A wide range of species use complex sounds to communicate, including humans and songbirds like zebra finches. During a critical period of learning, infants and young animals learn how to remember and discriminate this 'language' from other sounds. However, the changes that happen in the brain during this learning period are not well understood.

The process of learning forms new connections between neurons in the brain and prunes away old connections. These connections, known as synapses, come in different types. Signals sent across excitatory synapses increase the activity of the receiving neuron, while signals sent across inhibitory synapses reduce neuron activity.

What happens to the synapses in the brain during the critical period? To find out, Huang et al. used electron microscopy to examine the brains of young zebra finches that either had never heard birdsong, or had just heard birdsong for the first time. A single day of hearing song dramatically shifted the balance of excitatory and inhibitory synapses in the main vocal control area of the young birds' brains. The number of excitatory synapses decreased, and the number of inhibitory synapses increased.

The balance between excitation and inhibition is important for the brain to work correctly. Therefore, as well as helping us to understand how infants learn their first language, the results presented by Huang et al. could also help us to improve treatments for conditions where this balance goes wrong, such as mood disorders. For example, tailoring the time point of medication intake in combination with sensory exposure therapies could improve how effectively either one works.

DOI: https://doi.org/10.7554/eLife.37571.002

participate in successful acquisition of a template of tutor song, because blockage of NMDA-mediated (excitatory) synaptic transmission in HVC during tutor exposure abolishes normal song development (*Roberts et al., 2012*). One important role of HVC inhibition may be to protect HVC neurons after tutoring from further sensory influence, as the strength of HVC inhibition correlates with the learned portions of song but not with age (*Vallentin et al., 2016*). To obtain structural insights into the dynamics of HVC excitation and inhibition, we study the HVC synaptic organization and the changes associated with tutor exposure and with aging.

## Results

### Experiment I: the effect of tutoring on synapse densities

Using electron microscopy (EM), we investigated the effects of tutor song exposure on HVC synapses. We compared synapse densities in zebra finches that were either exposed to an adult tutor for one day (SHORT birds) or that were never exposed to a tutor (ISO birds). To compare with densities in normally reared birds, we included a third group of birds that were tutored for 24 days (LONG birds). In this first experiment, we minimized the influence of age by sacrificing all birds mid-development at 59 days post hatch (dph), which is near the end of the critical sensory song learning period, *Table 1*, *Figure 1A*.

We found that one day of tutor exposure led to a 26 ± 5% decrease in the HVC asymmetric (excitatory) synapse density (p = 3*10$^{-9}$, linear mixed effect (LME) model with bird group as fixed effect and bird identity as random effect, df = 1152, n = 4 SHORT birds and n = 4 ISO birds, see Materials and methods), *Figure 1B,C*. In contrast, the HVC symmetric (inhibitory) synapse density increased by about 20 ± 10% during one day of tutor exposure (p = 0.05, LME model, df = 1152, n=4 SHORT and n=4 ISO birds), *Figure 1B,D*. In combination, upon first tutoring and within 24 hr, the percent inhibitory synapses in HVC rose by 42% from on average 19% (ISO group) to 27% (SHORT group, p = 1.7*10$^{-6}$, LME model with 32 observations, two fixed effect coefficients, and eight random effect coefficients, see Materials and methods), *Figure 1E*.

Extended tutoring (LONG) had the net effect of removing both excitatory and inhibitory synapses. Compared with the ISO group, we found that 24 days of tutoring led to a decrease in excitatory synapse density of 37 ± 1% (p = 4.5*10$^{-18}$, LME model, n = 4 LONG and n = 4 ISO birds) and to a

**Table 1.** Bird groups used in Experiments I (ISO, SHORT, and LONG) and in Experiment II (ISO30, ISO60, ISO90, LONG60, LONG90)

| Bird group | Tutoring (dph) | Tutoring H/day | Sacrifice (dph) |
|---|---|---|---|
| ISO | - | - | 59 |
| SHORT | 58 | 24 | 59 |
| LONG | 35–58 | 24 | 59 |
| ISO30 | - | | 60 |
| ISO60 | - | - | 60 |
| ISO90 | - | - | 90 |
| LONG60 | 35–59 | 2 | 60 |
| LONG90 | 35–39 | 2 | 90 |

DOI: https://doi.org/10.7554/eLife.37571.010

decrease in inhibitory synapse density of 30 ± 2% (p = 0.003, LME model, n = 4 LONG and n = 4 ISO birds). Altogether, tutoring induced a synaptic bias towards inhibition that was transient, because the percent inhibitory synapses in LONG birds was barely larger than in ISO birds (21% vs 20%, p = 0.48, LME model, n = 4 LONG and n = 4 ISO birds).

## Experiment II: the effects of age and tutoring on synapse densities

To assess whether the observed changes in synapse densities were aligned (or anti-aligned) with normal developmental trends, in a second experiment, we studied HVC synapse turnover and the excitatory-inhibitory balance as a function of age. We measured HVC synapse densities at the onset of the sensory learning phase at 30 dph in untutored birds (ISO30 birds) and at the onset of adulthood at 90 dph, both in extensively tutored birds (LONG90 birds) and in untutored birds (ISO90 birds). We included two additional groups of extensively tutored and untutored birds sacrificed mid-development (60 dph, LONG60 and ISO60 birds), *Table 1*, *Figure 2A*.

We provided tutored (LONG60 and LONG90) birds in this second experiment with just 90 min of tutor exposure per day instead of the unrestricted exposure we had provided to LONG birds in the first experiment. The motivation for this change in tutoring paradigm was to induce better song copying. Song learning in juvenile zebra finches is inversely related to tutor song abundance, and more playbacks of a given tutor song lead to less complete song imitations (*Tchernichovski et al., 1999*). LONG birds in the first experiment produced rather poor song imitations, at 59 dph their average similarity score with tutor song was only about 27%. As expected, LONG60 birds tutored in 24 sessions of 90 min produced more accurate song copies: their average similarity score with tutor song exceeded that of LONG birds by 21% on average (p = 0.006, nonzero fixed effect of 0.21 in LME model comparing n = 4 LONG60 with n = 4 LONG birds, t = 2.41, df = 158), *Figure 2B*.

Extended tutoring was again observed together with synapse removal: at 60 dph, both excitatory and inhibitory synapse densities were lower in tutored than in untutored birds (p = $10^{-30}$ for excitatory density and p = $3*10^{-6}$ for inhibitory density, LME model, n = 4 LONG60 and n = 4 ISO60 birds). At 90 dph, densities remained lower in tutored birds compared to their age-matched untutored controls (p = 0.02 for excitatory density and p = $3*10^{-5}$ for inhibitory density, LME model, n = 4 LONG90 and n = 4 ISO90 birds), *Figure 2C,D*. Sparse tutoring was associated with lower excitatory synapse densities than dense tutoring (p = $3*10^{-6}$, LME model, n = 4 LONG60 and n = 4 LONG birds), suggesting that pruning of excitatory synapses is related to song maturation (i.e., increases in song similarity scores). Indeed, there was a negative correlation between song similarity scores and excitatory synapse densities (r = −0.77, p = 0.026, Pearson correlation coefficient, n = 4 LONG and n = 4 LONG60 birds), *Figure 2E*. Sparse tutoring did not appear to alter inhibitory synapse densities at 60 dph (p = 0.2, LME model, n = 4 LONG60 and n = 4 LONG birds).

In tutored birds, as a function of age, excitatory synapse densities followed a trough-increase trend (*Figure 2C*), whereas inhibitory synapse densities followed a mirrored peak-decline trend (*Figure 2D*). In tutored birds from 60 to 90 dph, excitatory synapse densities increased (p = $9*10^{-6}$,

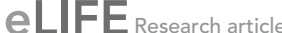

**Figure 1.** Effect of tutoring on HVC synapse densities, experiment I. (**A**) Experiment timeline. Male zebra finches were either tutored for one day (SHORT), for 24 days (LONG), or not at all (ISO), before being sacrificed at 59 days post hatch (dph). (**B**) We identified excitatory and inhibitory synapses in HVC tissue imaged with serial section electron microscopy (ssEM). (**C**) The excitatory synapse density decreases with increasing tutor exposure. (**D**) The inhibitory synapse density increases in briefly tutored birds, but it decreases in chronically tutored birds. (**E**) Tutoring induces a transient increase in the percentage of inhibitory synapses. (**C-E**) White/gray bars represent group means and the error bars represent the means in individual birds ± the standard deviations.

DOI: https://doi.org/10.7554/eLife.37571.003

The following figure supplements are available for figure 1:

**Figure supplement 1.** HVC volume in Experiment II.
DOI: https://doi.org/10.7554/eLife.37571.004

**Figure supplement 2.** Estimated tissue deformation $R_e$ caused by EM embedding, reported for each of the 32 samples.
DOI: https://doi.org/10.7554/eLife.37571.005

**Figure supplement 3.** Estimation of tissue deformation caused by ultramicrotomy.
DOI: https://doi.org/10.7554/eLife.37571.006

**Figure supplement 4.** Thickness estimation of ultrathin sections using cylindrical objects.
DOI: https://doi.org/10.7554/eLife.37571.007

**Figure supplement 5.** Tissue deformations $R_X$ (blue), $R_Y$ (red), $R_z$ (green) caused by ultramicrotomy.
DOI: https://doi.org/10.7554/eLife.37571.008

**Figure supplement 6.** Cumulative tissue deformation caused by EM embedding and ultramicrotomy, seperately reported for each sample.
DOI: https://doi.org/10.7554/eLife.37571.009

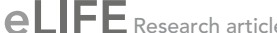

**Figure 2.** Effect of aging on HVC synapse densities, experiment II. (**A**) Experiment timeline. Male zebra finches were either tutored for 24 days (LONG), or not at all (ISO). Birds were sacrificed at 30, 60, or 90 dph. (**B**) Sparsely tutored (LONG60) birds produced good song copies at 60 dph, their similarity score with tutor song was higher on average than in densely tutored (LONG) birds. (**C**) Bar plots of excitatory and **D**) of inhibitory synapse densities. B-D White/gray bars represent group means and the error bars represent means in individual birds ± the standard deviations. (**E**) At 60 dph, there was a negative correlation between song similarity and excitatory synapse density, r = −0.77, p = 0.026.
DOI: https://doi.org/10.7554/eLife.37571.011

LME model, n = 4 LONG60 and n = 4 LONG90 birds) whereas inhibitory synapse densities decreased (p = 0.03, LME model, n = 4 LONG60 and n = 4 LONG90 birds). As a consequence, over the course of song development, the percentage of inhibitory synapses in tutored birds ranged from 7% at the onset of song development (30 dph), to roughly 23% at 60 dph, down to 14% at 90 dph. Thus, the developmental trajectory of the synaptic balance is highly dynamic in normally tutored birds.

In untutored birds, synaptic densities did not stay constant, either. The age-related peak-decline trend of inhibitory synapse densities occurred irrespective of sensory experience (it was seen in both tutored and untutored birds, *Figure 2D*) By contrast, in untutored birds, excitatory synapse densities did not decrease until past closure of the critical period, *Figure 2C*. At the young age of 30 dph, birds normally do not sing yet, except that they occasionally may produce highly unstructured sub-songs that do not require HVC (*Aronov et al., 2008*). These observations suggest that abundance of HVC excitatory synapses in young birds is not a requirement for song production, but that instead it signals readiness for sensory memory formation. Moreover, in untutored birds from 60 to 90 dph, both excitatory and inhibitory synapse densities decreased (p = 0.009 for excitatory and p = $3*10^{-6}$ for inhibitory synapses, LME model, n = 4 ISO60 and n = 4 ISO90 birds), suggesting that high synapse densities are costly and therefore not maintained beyond the critical period.

We combined the data from both experiments to perform a group-level analysis on the (near) 60 dph animals. Combining the ISO and ISO60 birds, we found that one day of tutoring was associated with a 22% increase in inhibitory synapse densities (p = 0.016, n = 8 ISO and n = 4 SHORT birds,

Wilcoxon ranksum test) and with a 33% decrease in excitatory synapse density (p = 0.008, n = 8 ISO and n = 4 SHORT birds, Wilcoxon ranksum test). In combination, one day of tutoring was associated with an increase in the percentage of inhibitory synapses from 20% to 29% (p = 0.004, n = 4 SHORT and n = 8 ISO birds, Wilcoxon ranksum test). Combining also the LONG and LONG60 birds, we found that extended tutoring was associated with a decrease in inhibitory and excitatory synapse densities of 38% and 50%, respectively (both tests p = 0.00016, n = 8 LONG and n = 8 ISO birds, Wilcoxon ranksum test), *Figure 3A,B*. Tutoring has no major influence on HVC size (*Bottjer et al., 1985*; *Herrmann and Bischof, 1986*; *Nordeen and Nordeen, 1988*). We therefore expected our findings on synapse densities to translate more or less directly into findings on synapse numbers. Indeed, we found that HVC volumes were variable across animals but did not strongly depend on age or treatment, with the exception that the average HVC volume in 30 dph-old birds was about 30% smaller than in the other bird groups (see *Figure 1—figure supplement 1*). Despite this moderate growth in HVC volume from 30 to 60 dph, LONG60 birds tended to have fewer excitatory HVC synapses than did ISO30 birds, evidencing in sparse tutored birds from early to mid-development a process of net excitatory synapse elimination.

## Effects of tutoring on synapse sizes

Synapse insertions and deletions are extreme cases of subtler processes of synapse size changes. To inspect the latter, we reconstructed HVC synapses in brains from the first experiment, imaged with focused ion beam electron microscopy (FIBSEM, see Materials and methods), *Figure 4A–C*. We found that synapse sizes (their physical volumes) were well approximated by log-normal distributions, *Figure 4—figure supplement 1*. Log mean synapse sizes displayed large differences across animals. Nevertheless, we found an effect of 40% larger excitatory synapses in SHORT birds compared to ISO birds (LME, p = 0.05), *Figure 4D*. Extended tutoring was not associated with significant size changes of excitatory synapses (LME, ISO vs LONG, p = 0.42). Sizes of inhibitory synapses were not affected by either short or long tutoring (LME, p > 0.05), *Figure 4E*. Also, we found no effect of tutoring on Feret diameters (directional measure of object size) of either excitatory or inhibitory synapses (LME, p > 0.05).

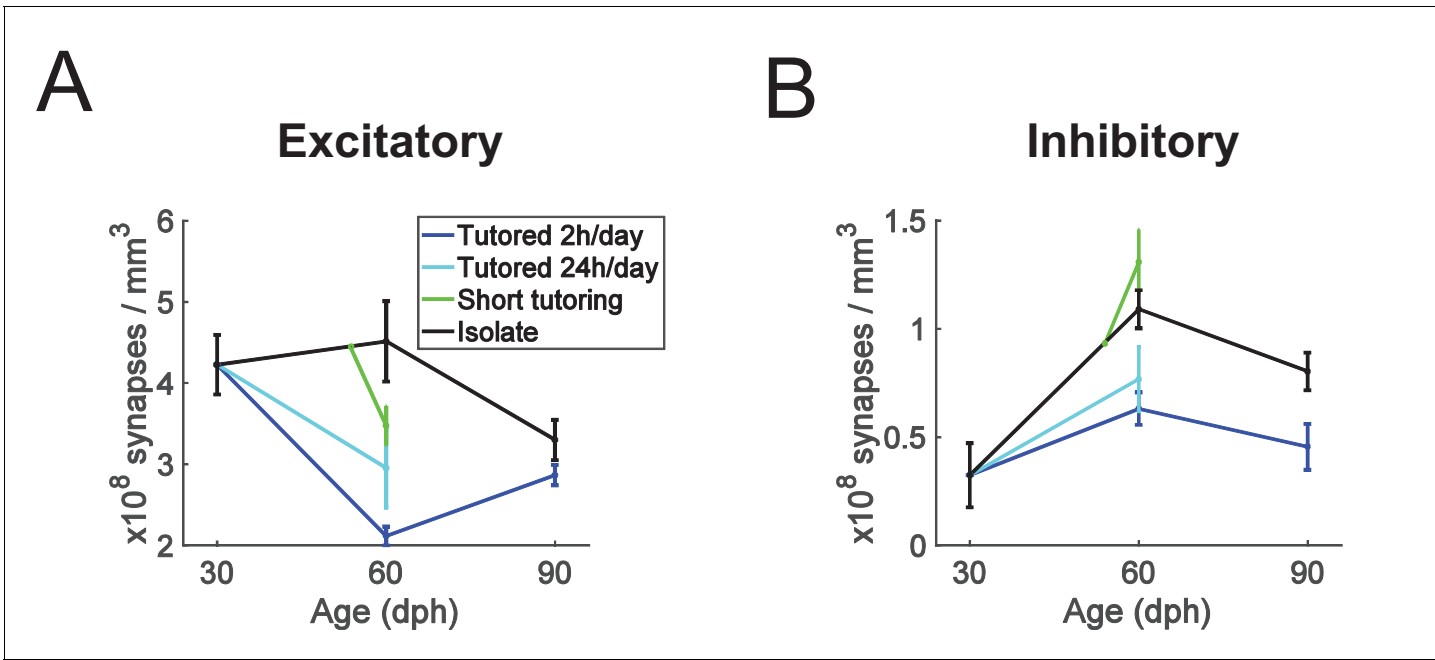

**Figure 3.** Summary plots showing age and experience dependence of excitatory and inhibitory synapse densities. (A) excitatory synapses, (B) inhibitory synapses. Error bars depict stand deviations.
DOI: https://doi.org/10.7554/eLife.37571.012

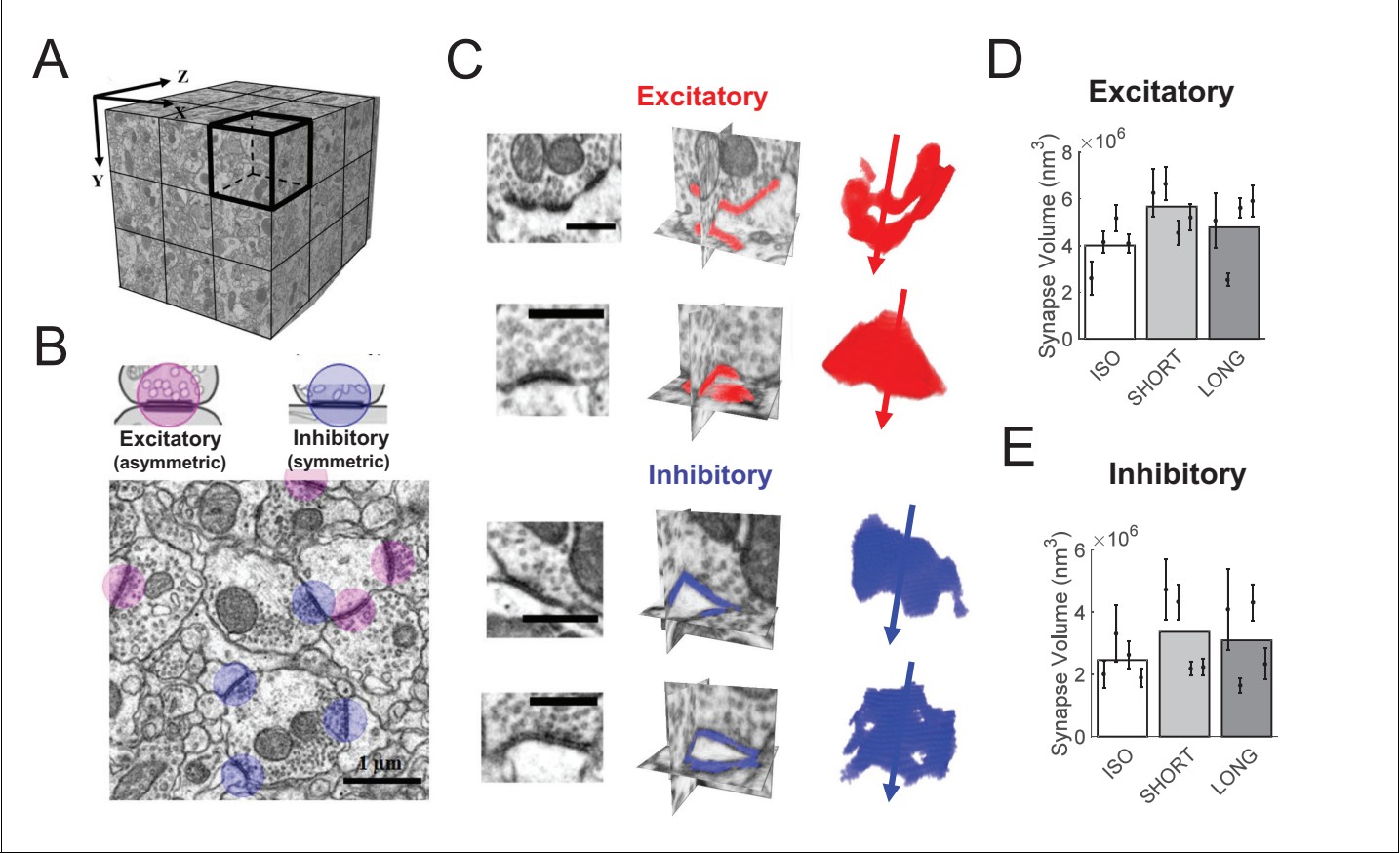

**Figure 4.** HVC synapse sizes are variable and weakly affected by tutoring. (**A**) We acquired close to isotropic imagery of an (8 μm)³ cube of HVC tissue using focused ion beam electron microscopy (FIBSEM). (**B**) Example image of excitatory (red) and inhibitory (blue) synapses. (**C**) Example reconstructed excitatory (red) and inhibitory (blue) synapses. Shown are the synapses in the original FIBSEM dataset (left), 3D orthoslices intersected at the synapse centers with the segmented synapse voxels shown in color (middle), and 3D reconstruction of the segmented voxels with arrows pointing to the postsynaptic side (right). Scale bars: 0.5 μm. (**D**) Excitatory (asymmetric) synapses in tutored (SHORT) birds were larger than in untutored (ISO) birds, no significant change in synapse size was associated with extensive tutoring (LONG birds). (**E**) Neither SHORT nor LONG tutoring was associated with significant changes in inhibitory synapse sizes. C-D White/gray bars represent group means and the error bars represent the means in individual birds ± the standard deviations.

DOI: https://doi.org/10.7554/eLife.37571.013

The following figure supplements are available for figure 4:

**Figure supplement 1.** Histograms of synapse sizes per bird group and synapse type.

DOI: https://doi.org/10.7554/eLife.37571.014

**Figure supplement 2.** Segmenting synapses in FIBSEM imagery using Ilastik.

DOI: https://doi.org/10.7554/eLife.37571.015

## Discussion

Our findings suggest that insertion and elimination of HVC synapses are age and experience dependent. There was a strong decoupling among synapse types in that we observed a more than three-fold variation across age and experience in the relative density of inhibitory synapses. After extended tutor exposure, the density, ratio, and average sizes of synapses returned to levels close to those of age-matched untutored controls, which hints at a structural homeostasis (*Turrigiano and Nelson, 2004*) that is independent of learning experience.

Synapse density has been commonly observed to follow a peak-decline trend from youth to adulthood (*Cragg, 1975*; *Herrmann and Arnold, 1991*; *Murphy and Magness, 1984*; *Winfield, 1981a*). The peak is thought to represent overproduction of neuronal connections, reflecting elevated structural plasticity at the onset of behavioral development. In HVC of normally raised

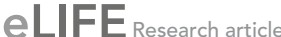

birds, we find an early peak-decline trend for excitatory synapses, and a late such trend in inhibitory synapses.

Excitatory synapses were formed very early and in untutored birds they were regulated at a constant rate throughout the sensory learning period from 30 to 60 dph. Inhibitory synapses were formed later, reminiscent of delayed development of inhibition in visual (*Winfield, 1981b*) and auditory cortices (*Dorrn et al., 2010*). Given that increases in inhibition tends to cause the end of experience-related plasticity (*Chen et al., 2011*; *Fagiolini and Hensch, 2000*; *Hensch, 2005*; *Iwai et al., 2003*), the age- and tutoring dependent increases in inhibitory synapse densities we find raise the possibility that increased inhibition causes not only the closure of the sensory learning period but also accelerates the ending of structural plasticity right after tutoring.

Processes of synapse formation and elimination are associated with sprouting and retraction of dendritic spines (*Holtmaat et al., 2006*; *Trachtenberg et al., 2002*). Because tutoring is associated with a rapid and monotonic decline in HVC spine turnover (*Roberts et al., 2010*), there seems to be diminished requirement for structural plasticity right after tutoring, in line with our observed increase in inhibitory synapse density that supposedly closes critical period learning.

Synapse pruning tends to be an activity-dependent process, for example mediated by microglia (*Schafer et al., 2012*). Tutor exposure is associated with decreases in excitatory synapse densities, and so naively one would expect higher excitatory synapse densities in sparsely tutored birds compared to densely tutored birds. However, we find the opposite, and therefore the better explanation for fewer excitatory synapses in sparsely tutored birds is their higher song maturity (better song imitation). The extent to which higher maturity stems either from better memorization of tutor song or from more targeted song practice remains to be investigated. Neither explanation can be currently ruled out, although our findings align mostly with the former explanation, as detailed in the following two paragraphs.

On the one hand, excitatory connections have been proposed to implement a sequence-generating chain network (*Fee et al., 2004*; *Kornfeld et al., 2017*). Elimination of HVC excitatory synapses during song maturation agrees with the sparse firing of excitatory motor-projecting neurons in adulthood (*Kozhevnikov and Fee, 2007*; *Hahnloser et al., 2002*). Over the course of song maturation, HVC projection neurons fire first densely during each syllable and then sparsely during only a single syllable type (*Okubo et al., 2015*). Such sparsening of firing has been suggested to result from a synaptic chain splitting mechanism that underlies the transformation of immature subsongs into more stereotyped plastic songs. Our findings of progressive elimination of excitatory synapses up to 60 dph supports such chain splitting, whereas the subsequent insertion of excitatory synapses (LONG90 vs LONG60) could reflect a process of chain strengthening.

On the other hand, excitatory synapses and their dendritic spines can be formed by repetitive motor training (*Xu et al., 2009*; *Fu et al., 2012*), which aligns with our observation of increasing excitatory synapse density towards adulthood. Given that optical ablation of task-specific spines in motor cortex can selectively disrupt newly acquired motor skills (*Hayashi-Takagi et al., 2015*), it is likely that the late-formed excitatory synapses between 60 and 90 dph contribute to increased motor performance in adults. Accordingly, our findings support the view that sensory memorization is associated with loss of excitatory synapses, whereas increases in motor performance are associated with insertion of excitatory synapses.

Paradoxically, we find a net removal of inhibitory synapses (LONG vs ISO) during a developmental phase in which the strength of inhibition onto motor-projecting HVC neurons increases (*Vallentin et al., 2016*). Likely, these contrasting findings can be reconciled by maturation processes such as insertion of AMPA receptors at excitatory synapses (*Hensch, 2005*), for which we find indirect evidence in terms of increase in excitatory synapse size (*Nusser et al., 1998*). Additionally, strengthened inhibition with fewer inhibitory synapses can arise when synapse elimination primarily affects competing neuron pools linked by inhibition (*Kornfeld et al., 2017*), that is when eliminated synapses disinhibit the local excitatory-inhibitory pools (*Markowitz et al., 2015*) that are associated with learned song syllables (*Vallentin et al., 2016*), *Figure 5*.

Removal of excitatory inputs is in line with developmental sharpening of excitatory inputs seen in auditory cortex (*Sun et al., 2010*). However, it is unclear how the structural preponderance of excitatory versus inhibitory synapses relates to the functional balance between excitatory and inhibitory synaptic inputs. Functional balances in individual neurons can be measured using whole-cell

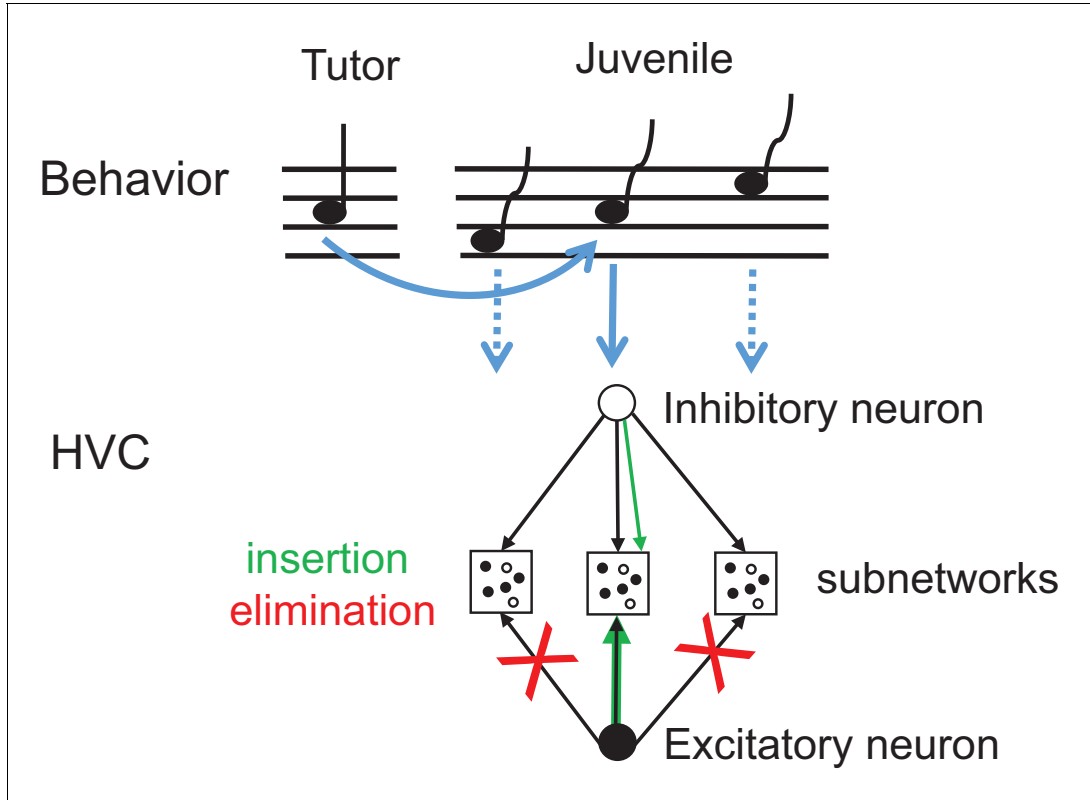

**Figure 5.** Synapse elimination model of tutor song memorization. Exposure to tutor song leads to insertion of those inhibitory synapses and strengthening of those excitatory synapses (green arrows) that are part of the HVC subnetwork (middle) involved in production of a note resembling a tutor note (full blue arrow). By contrast, excitatory synapses associated with notes and note transitions unlike in tutor song (dashed blue arrows) are eliminated (red crosses) right after tutoring. The HVC subnetworks associated with diverse notes in the juvenile's song need not to be topographically organized as shown here.

DOI: https://doi.org/10.7554/eLife.37571.016

recordings, but it will be excessively difficult to obtain such information from singing birds, given that such recordings have not been feasible thus far.

Overall, our work provides a structural foundation of developmental learning mechanisms and the roles of neurotransmitter systems. Given that the pathophysiology and treatment of many severe mood disorders are linked with excitatory (glutamatergic) and inhibitory (GABAergic) neurotransmitters, imbalances in these systems either associated with depression or induced by antidepressants (*Sanacora et al., 2003*) can interfere with requirements dictated by age and experience. For example, depression during pregnancy extends the critical period for infant speech discrimination, whereas treatment of depression with selective serotonin reuptake inhibitors (SSRIs) shortens this critical period (*Weikum et al., 2012*). SSRI-induced shortening of critical periods might be caused by physiological deficits such as premature increases in inhibition (*Sanacora et al., 2002*), premature decreases in excitation, or a combination of both mechanisms, given their known interdependence (*Nakayama et al., 2012*). The roles of amino acid neurotransmitters in depression and mood disorders appear diverse because there have been mixed reports of SSRIs causing either increases or decreases in cortical GABA and glutamate concentration (*Maya Vetencourt et al., 2008*; *Sanacora et al., 2002*; *Sanacora et al., 2003*). Part of the conundrum might be the alternating experience-dependent dynamics in inhibitory and excitatory synapses we find. Our work suggests that exposure therapies and the pharmacological treatment of mood disorders might benefit from targeted manipulation of both excitation and inhibition, taking into consideration their rich experience-dependent dynamics.

One expected immediate effect of a tutoring-induced bias towards inhibition is to reduce neural activity in postsynaptic targets. However, more exciting is the idea that our observed structural

changes are associated with holding a memory of the tutor song. On the one hand, a memory could be held by inhibitory synapses inserted right after tutoring. Support for this idea comes from barrel cortex, where single whisker stimulation triggers formation of inhibitory synapses within 24 hr in the corresponding barrel but not in other barrels (*Knott et al., 2002*). On the other hand, a song memory could be formed by irreversible removal of excitatory synapses right after tutoring. Accordingly, the tutor template would be stored by pruning unused excitatory connections, *Figure 5*. In this model, the initially formed excitatory connections would represent the entire hypothetical space of notes and note transitions, consistent with zebra finches' vocal repertoire. The eliminated excitatory connections would represent the notes and note transitions that are not matched by tutor song, whereas the strengthened synapses would be the ones required for tutor song imitation.

## Materials and methods

### Experimental subjects

34 male juvenile zebra finches (<90 days) were raised in our facility at the University of Zurich, Switzerland. At 15 days post hatching (dph), the young birds were transferred together with their mothers to a sound-isolation chamber where they could not hear male songs. Between 25 and 35 dph, sex determination was performed based on feather appearance and on genotyping. At 35 dph, selected male birds were separated from siblings and transferred to individual sound-isolation chambers (inner volume $60 \times 60 \times 60$ cm$^3$) where their songs were recorded.

We randomly assigned birds to treatment groups with one exception: to minimize the influence of genetic background on results, siblings from the same nest were never put in the same group.

Birds were housed in two cages with inner dimensions $39 \times 23 \times 39$ cm$^3$ (length ×width × height), joined together by the doors, resulting in a movement range of approximately $39 \times 46 \times 39$ cm$^3$. Each juvenile bird was housed together in the same cage with an adult bird, inside the recording chamber: either a male serving as live tutor or a female serving as companion (female zebra finches do not sing). Singing of juveniles and tutors was recorded continuously and monitored on a daily basis. We aborted the experiment in two juveniles because the tutor failed to sing at least 20 song motifs per day. Birds were maintained on a 14:10 hr light:dark cycle with food and water provided ad libitum. To avoid human bias, we coded all birds with numeric identifiers and performed all analyses in a manner that was blind to bird identity and treatment group. All experimental procedures were in accordance with the Veterinary Office of the Canton of Zurich.

### Experiment I

Experiment I was designed to investigate whether experience of the tutor song alters excitatory and inhibitory synaptic connectivity in HVC. We divided birds into the following three groups (*Figure 1A*), each composed of four animals:

- Fully isolated group (ISO): From 35 to 59 dph on, each juvenile was housed together with a female companion and completely isolated from zebra finch song.
- Short-tutored group (SHORT): From 35 to 58 dph, each juvenile was housed together with a female companion and completely isolated from zebra finch song. At 58 dph, 30–60 min after the lights turned on, the female was replaced by a tutor. At 59 dph, the juvenile was sacrificed, exactly 24 hr after introduction of the tutor.
- Long-tutored group (LONG): From 35 dph on, each juvenile was housed together with a tutor.

At 59 dph, all juveniles were deeply anesthetized and perfused.

### Experiment II

Experiment II was designed to explore changes in HVC excitatory and inhibitory synapses as a function of age. To separately explore age dependence in tutored and untutored birds, we divided birds into the following five groups, each composed of four animals:

- 30 dph isolated group (ISO30): Each juvenile zebra finch was raised by an adult female and separated from its siblings at $27 \pm 1$ dph, after which it was transferred to a sound isolation chamber and provided an adult female as social companion. When it reached 30 dph, it was sacrificed in the morning and its brain was extracted.

- 60 dph isolated group (ISO60): Each juvenile was given the same treatment as ISO birds in Experiment I. This biological replicate allowed us to increase the size of the ISO group and to perform population-level analyses (Wilcoxon ranksum tests).
- 60 dph tutored group (LONG60): Each juvenile was raised by an adult female and separated from its siblings at 35 dph. Thereafter, the juvenile was transferred to an individual recording chamber, where it was provided social company by an adult female zebra finch. From 35 dph to 59 dph, a live tutor was presented to the juvenile every morning for a duration of 90–120 min. The tutor was transferred to a presentation cage of dimensions $35 \times 16 \times 29$ cm$^3$ at least one hour before being presented to the juvenile. In the presentation cage, the tutor had access to food, water, and enrichment tools, but physical contact with the juvenile was not possible. When the juvenile reached 60 dph, it was sacrificed in the morning.
- 90 dph isolated group (ISO90): Each juvenile in this group was treated like juveniles in the ISO60 group, except that it was housed together with a female until 90 dph (instead of 60 dph), when it was sacrificed.
- 90 dph tutored group (LONG90): Each juvenile in this group was treated until 60 dph like juveniles in the LONG60 group. From 60 to 90 dph, each bird was housed together with a female companion. At 90 dph, the bird was sacrificed.

## Song analysis

All vocalizations produced by the juveniles and their companions were recorded with a wall-mounted microphone (Audio Technica PRO 42), amplified with a microphone preamplifier (RME Quadmic), and digitally sampled at 32 kHz (PCI card, National Instruments). Songs were detected and saved using custom written software (Labview, National Instruments).

To evaluate the similarity between the juveniles' songs and tutors' songs one day before sacrifice, we randomly selected from the last day of recording 20 song motifs from the pupil and 20 motifs from the tutor. The comparison of the 20 motif pairings was performed using Sound Analysis Pro (SAP) (*Tchernichovski et al., 2000*).

## Brain sample collection, fixation, and sectioning

Birds were sacrificed with 50 µL sodium pentobarbital (Streuli Pharma AG Esconarkon) injected intramuscularly. They were perfused with 5 mL 0.9% NaCl, followed by 300 ml freshly prepared fixative solution at body temperature (4% paraformaldehyde and 0.1% glutaraldehyde in 0.1 M pH 7.4 phosphate buffer (PB)). Our procedure was adapted from *Knott et al. (2011)*, instead we used a slightly reduced glutaraldehyde concentration. After perfusion, brains were dissected from the skull and briefly washed in PB. Prior to sectioning, to enhance cutting stability, we separated the two hemispheres along the midline and embedded each in 3% agar. Parasagittal brain slices of 100 µm thickness were cut on a vibratome (Thermo Scientific, Microm HM 650V) and collected at 4°C in 0.1 M PB.

The sections were first washed in cacodylate buffer (0.1 M, pH 7.4) and then incubated for 40 min in 1.5% potassium ferrocyanide and 1% OsO4 in cacodylate buffer, following another 40 min incubation in only 1% OsO4. Thereafter, the sections were incubated for another 40 min in 1% uranyl acetate in double distilled water (ddH2O). After these heavy metal stainings, the sections were dehydrated by 10 min incubations each in a gradient of ethanol solutions with increasing concentrations (50%, 70%, 90%, and 95%). The sections were further dehydrated by twice incubating in 100% ethanol following twice incubating in propylene oxide, each for 15 min. During the dehydration, small amounts of liquid from the preceding steps remained inside the vial to prevent the tissue from drying and cracking. After dehydration, the sections were immersed into a freshly made epoxy resin for at least 12 hr to reach complete tissue infiltration. The epoxy resin (Durcupan ACM, Sigma-Aldrich Fine Chemicals, Buchs, Switzerland) consisted of 10 g: 10 g: 0.3 g: 0.2 g of component A/M, B, C, and D, respectively. After incubation, the sections were placed between two Aclar films (ACLAR sheets, Agar Scientific Ltd., Stansted, UK) that were sandwiched between two glass slides. A small weight (~40 g) was placed on top of the upper glass slide. The sections were then cured for 48 hr at 52°C.

Before proceeding to ultramicrotomy, the ROI in HVC was located by comparing the LM images of a section before and after embedding. With a scalpel blade (cat. # 10050–00; Fine Science Tools GmbH., Heidelberg, Germany), a small piece of tissue containing HVC was dissected. The small

tissue piece was then glued on top of a resin block with the brain tissue side facing down (remaining Aclar film facing up).

HVC was located by reference to the LM images and trimmed in a trapezoidal shape with a diamond trimming knife (trimtool 20, DiATOME Ltd., Nidau, Switzerland). The Aclar film on top was trimmed away and the surface of the brain tissue block was polished. Serial 70-nm-thick ultrathin sections were cut with an ultramicrotome (Leica FC6) and a sharp diamond knife (Histo Jumbo, DiATOME Ltd). The sections formed a ribbon of tissue that were floating on the surface of the water bath. The section thickness was monitored based on the reflection index (golden:>100 nm; silver-gray: 50–100 nm). Sections that were too thick were discarded from collection. When the desired ribbon of serial ultrathin sections was produced, it was detached from the diamond blade and moved on the water surface with a human eyelash that was fixed to a toothpick. A custom-made silicon wafer (Si-Mat Silicon Materials, Kaufering, Germany) was first deionized with a charging generator (EN SL, Haug Biel AG, Biel, Switzerland) and then slowly dipped into the water bath. The floating ribbon of ultra-thin sections was flattened and attached to the silicon wafer and both were withdrawn from the water bath. Prior to the ssSEM imaging of the dried ultrathin sections, the silicon wafer was fixed with clips onto a SEM sample holder (cat. # 16112–20, Plano GmbH, Wetzlar, Germany).

## HVC size

HVC volume grows significantly from 20 to 40 dph, after which it reaches 91% of its adult size (*Bottjer et al., 1985*;*Herrmann and Bischof, 1986*;*Nordeen and Nordeen, 1988*). The size of HVC is mainly genetically regulated (*Airey and DeVoogd, 2000a*) and manipulations of song experience have little influence (*Brenowitz et al., 1995*; *Burek et al., 1991*). However, although HVC volume and neuron number remain constant in adults (*Wang et al., 2002*), both HVC volume and HVC neuron number positively correlate with the number of song syllables copied from a tutor (*Airey et al., 2000b*; *Ward et al., 1998*).

To measure HVC size in Experiment II, wet brain sections and Nissl-stained brain sections were imaged under a bright field light microscope (Olympus BX61). Wet brain sections were first mounted onto glass slides and kept moist with PB during imaging. Light microscopy images of HVC were taken at different magnifications (1.25x, 4x, and 10x). To compensate for fluctuations in environmental illumination, microscope parameters were set prior to imaging such that histograms of live images appeared normalized.

In order to measure HVC size defined as the physical volume of HVC, we adopted the procedure described by Airey and colleagues (*Airey et al., 2000c*). Basically, all images of HVC in a given hemisphere were grouped together and imported as an image stack into a single TrakEM2 project. The images were then aligned using translational and rotational transformations. The HVC outline on each image was hand-drawn and the number $n$ of pixels inside HVC was calculated. With known pixel size $p$ and section thickness $d$, the size $V_{sec}$ of HVC in a given brain section was estimated as the product of these three terms: $V_{sec} = npd$. The total size $V_{HVC}$ was calculated as the sum over all sections: $V_{HVC} = \sum_{sec} V_{sec}$ and the final HVC size for a given animal was calculated as the average of the right and left HVC sizes,*Figure 1—figure supplement 1*.

To test for tutoring or aging-related changes in HVC size, we modeled HVC sizes using a linear mixed effects (LME) model with bird group as fixed effect and bird identity as random effects (see Section on Linear Mixed Effects Models). We found the following peak-decline trend: at 30 dph, HVC size was smaller than at 60 dph (p = 0.0004, LME model, ISO30 and ISO60); and at 60 dph, HVC size was larger than at 90 dph (p = 0.02, LME model, ISO60 and ISO90). Furthermore, we found that extended tutoring lead to HVC shrinkage at 60 dph (p = 0.03, ISO60 vs LONG60), whereas no effect of tutoring on HVC size was observed at 90 dph (p > 0.05, LME model ISO90 vs TUT90), which agrees with previous reports (*Brenowitz et al., 1995*; *Burek et al., 1991*).

## Serial section electron microscopy (ssEM)

For synapse counting, we performed serial section electron microscopy (ssSEM) on thin tissue volumes of dimensions ~ 80 µm×80 µm×140 nm located near the center of HVC. We performed EM imaging with a high-throughput scanning electron microscope (Merlin, Carl Zeiss Microscopy GmbH, Oberkochen, Germany). The section holder was fixed onto the sample stage and loaded into the

microscope chamber. The approximate locations of the serial sections were first determined with 5 kV of extra-high tension (EHT, the voltage applied to the electron gun) and an in-lens detector. The subsequent working distance (the distance between the sample surface and the electron gun) was 3.5 mm. The EHT was then reduced to 1.6 ± 0.1 kV, and the view mode was switched to the energy selective backscattered electron detector (EsB), with the EsB Grid voltage set to 550 V and the detector probe current set to 550 pA. All high-resolution ssSEM datasets were acquired with the EsB detector in the pixel-averaged noise-reducing scanning mode. After fine adjustments of the imaging parameters, the focus and astigmatism correction were optimized. All of these adjustments were set manually with the SmartSEM software (Carl Zeiss Microscopy GmbH). We acquired images with a pixel size of 4 nm and an electron beam dwell time of 7 μs. Image acquisition was performed automatically using the ATLAS 3D software (Fibics Incorporated, Ottawa, Canada).

We identified excitatory and inhibitory synapses in EM imagery based on morphology (*Gray, 1969*; *Klemann and Roubos, 2011*). Excitatory (asymmetric) synapses typically display a pronounced post-synaptic density (PSD). In contrast, the PSD in inhibitory (symmetric) synapses looks similar to the presynaptic membrane, showing no obvious differences in membrane specialization. In addition, asymmetric synapses have wider synaptic clefts and are always associated with larger (~40 nm) and rounder synaptic vesicles when compared with symmetric synapses whose synaptic vesicles are smaller (~20 nm) and of irregular oval shape, *Figures 1B* and *4B*.

## Physical section deformation

In both Experiments I and II, physical disector volumes were carefully calibrated before estimating synapse densities. To quantify tissue deformations that can occur during EM staining, embedding, and ultramicrotomy, we took LM and EM images at diverse tissue preparation stages. Deformations resulting from tissue embedding were estimated by comparing LM images of wet brain sections taken prior to resin embedding with LM images of the same sections after embedding. In such image pairs, we identified landmarks such as blood vessels, ventricles, and sharp sample borders. We then estimated the distances $S$ between three pairs of such identified landmarks using Fiji.

We assumed that tissue deformations caused by EM staining and embedding were isotropic. We calculated the relative 3D physical volume deformation $R_e$ (shrinkage or dilation) from the ratio of measured 1-D line distances as $R_e = \left( \frac{\text{S(embedded section)}}{\text{S(wet section)}} \right)^3$. For each animal, we estimated the relative 3D physical volume deformation $R_e$ (shrinkage or dilation) caused by tissue embedding, *Figure 1—figure supplement 2*. We found that embedding caused a dilation of 2.2 ± 0.2 % in each spatial direction.

We similarly estimated tissue deformations caused by ultramicrotomy by comparing LM images of trimmed sample block faces with SEM images of ultrathin sections cut from the same sample block, *Figure 1—figure supplement 3*. Physical cutting was always along the same direction and thus introduced non-isotropic tissue deformations. Therefore, we separately measured the deformations along the three orthogonal axes X, Y, and Z. We defined the X-axis to be perpendicular to the cutting direction and the Y-axis to be parallel to the cutting direction (*Figure 1—figure supplement 3*, red and blue lines). We calculated the deformation $R_X$ along the X axis as the deformation ratio: $R_X = \frac{\text{S(ultrathin section)}}{\text{S(embedded section)}}$, the deformation $R_Y$ along the y axis was computed analogously. In summary, ultramicrotomy caused on average a 1.4 ± 0.3 % tissue dilation along the X-axis and a 17.9 ± 0.3 % shrinkage along the Y-axis (cutting direction).

Thicknesses of ultrathin sections along the Z-axis were estimated using a cylindrical diameter method adopted from Fiala and Harris (*Fiala and Harris, 2001*). This method provides an estimation of the average separation of a given set of consecutive 2D images based on precise measurements of the diameters of cylindrical objects such as mitochondria, *Figure 1—figure supplement 4*. For a mitochondrion $i$ that was longitudinally dissected in the images, we measured the diameter $d_i$ and counted the number $s_i$ of sections it spanned. By averaging the resulting ratio $\frac{d_i}{s_i}$ over all $N$ inspected mitochondria ($N \simeq 20$ in each bird), we obtained the following estimate $\bar{t}$ of mean section thickness: $\bar{t} = \frac{1}{N} \sum i \frac{d_i}{s_i}$. The average ultrathin section thickness was 1.5 ± 0.7 % thinner than the target thickness (70 nm) set in the ultra-microtome. The deformation $R_Z$ along the Z-axis was estimated as $R_Z = \frac{\bar{t}}{70 \text{ nm}}$, where 70 nm represents the advancement of the resin block between two consecutive

ultrathin section cuts. The deformations $R_Z$ of all samples are depicted as green bars in *Figure 1—figure supplement 5*; as expected, the tissue deformations caused by ultramicrotomy were non-isotropic, *Figure 1—figure supplement 5*. Along the cutting direction, the ultrathin sections shrunk to roughly 80% of their original size (Y-axis), whereas along the perpendicular X-axis there was essentially no deformation.

The cumulative tissue deformation $R_{cum}$ from wet brain sections to embedded ultrathin sections was estimated as $R_{cum} = R_e * R_X * R_Y * R_Z$, *Figure 1—figure supplement 6*. This number represents the overall change in tissue volume from the wet state to the ultrathin section state.

We calibrated density estimates for each sample with the value of $R_{cum}$. The synapse density estimates in Experiments I and II therefore reflected the density in the wet state, which was close to the in-vivo state. On average, physical volumes in ultrathin sections were 16.1 ± 0.7 % smaller than in wet brain sections.

## Disector counting

To estimate synapse densities, we used disector counting in ssSEM section pairs, which is a standard stereology method, *Figure 6*. Disector methods for synapse counting do not rely on the counting of every synapse in the sample. Disectors provide reliable estimates of object numbers (errors smaller than 6%) when the separating distance between consecutive disectors is no larger than twice the

**Figure 6.** The disector method for counting synapses in ssSEM datasets. (a) Ribbons of serial ultrathin brain sections were scanned with an electron microscope. (b) A pair of aligned sections were chosen as reference (left) and look-up (right) sections. A 5 × 5 µm grid (yellow) was superimposed onto corresponding regions of the reference and look-up sections to delineate the boundaries of the disectors. (c) In this zoom-in, the green lines delineate the inclusion boundaries, and the red lines delineate the exclusion boundaries. Dissected, excluded, and non-dissected example synapses are color-circled in green, red, and blue, respectively.

DOI: https://doi.org/10.7554/eLife.37571.017

mean size of objects. When the separating distance is larger than three times the mean object size, density errors rapidly increase to 27% (**Merchán-Pérez et al., 2009**). Density estimates converge to the true density when more than a hundred disectors are inspected (**Merchán-Pérez et al., 2009**).

In each animal, we calibrated disector physical volumes. In experiment I, we inspected on average 96 disectors per animal (range 68–111 disectors, n = 12 birds). Therein, we counted on average 188 dissected synapses per bird (range 120–245, n = 12 birds), among which on average 42 were symmetric (range 26–68, n = 12 birds). In experiment II, we inspected on average 102 disectors per animal (range 77–124 disectors, n = 20 birds). Therein, we counted on average 175 synapses per bird (range 114–250, n = 20 birds), among which on average 29 were symmetric (range 5–49, n = 20 birds). We found HVC synapse densities in the range 5–8 $\times$ $10^8$/mm$^3$, in consistency with previous EM studies of HVC (**Herrmann and Bischof, 1986**; **Peng et al., 2012**).

Synapse density estimates from ssSEM imagery could in principle be biased because of missed synapses located between two consecutive sections (e.g. synapses smaller than 70 nm oriented parallel to the imaging plane). However, based on our measured synapse sizes of around 200 nm, we expect that our density estimates are not severely affected by such biases (see Section on Synapse sizes).

## Estimation of percent symmetric synapses

To calculate the percentage of symmetric synapses, we computed in each disector $i$ the total number $N_i^s$ of dissected symmetric synapses and the total number $N_i$ of dissected synapses. The percentage $q$ of symmetric synapses can be estimated by dividing the sums of these two numbers, $q = 100 \frac{\sum_i^n N_i^s}{\sum_i^n N_i}$, where $n$ is the number of disectors. To estimate the variance of the percentage of symmetric synapses, we used a jackknife resampling procedure. Accordingly, the $i$ th estimate $q_i$ of symmetric synapse percentage was defined as:

$$q_i = 100 \frac{\sum_{k \neq i}^n N_i^s}{\sum_{k \neq i}^n N_i}.$$

The jackknife estimate $\bar{q}$ of mean percentage and the estimate $\sigma_q^2$ of its variance was then calculated as $\bar{q} = \frac{1}{n} \sum_i^n q_i = q$ and $\sigma_q^2 = \frac{n-1}{n} \sum_{i=1}^n (q_i - \bar{q})^2$, see (**Efron and Stein, 1981**; **Shao and Wu, 1989**). The mean and variance of symmetric synapse percentage are shown for each bird in **Figure 1E**.

We assessed statistical differences in the percentage of inhibitory synapses among the diverse bird groups in experiment 1 using two independent procedures. Our first approach was to use the method reported in the Results Section based on dividing the disectors in each animal into four separate groups and using an LME model on the 4 independent estimates of percent inhibitory synapses per animal. Our second approach was to use a basic bootstrapping procedure, in which we simulated $n = 10^6$ - 5 experiments with 4 ISO and 4 SHORT birds with Gaussian-distributed percent symmetric synapses, taking the per-animal statistics reported in **Figure 1E**. Using this latter bootstrapping procedure, we obtained a p-value of p = $5*10^{-5}$, where the p-value represents the fraction of the $10^6$ - 5 simulated experiments in which SHORT birds had a lower percentage of inhibitory synapses than ISO birds. Clearly, both approaches yielded the nearly identical statistical result that SHORT birds had a higher percentage of inhibitory synapses than ISO birds.

## FIBSEM imaging

We imaged carbon-coated brain sample blocks with a FIBSEM microscope (Auriga 40, Carl Zeiss Microscopy GmbH). We first set the working distance to at least 10 mm and obtained live images of the block surface under an EHT of 5 kV and using the secondary electron detector. Thereafter, the working distance was reduced to 4.8 mm, and the EHT was reduced to 1.7 ± 0.2 kV. We ensured that the ROI on the sample surface was located precisely at the coincident point of the cross-beam system of the instrument. We switched the imaging modes back and forth between the SEM and FIB views. The Z-axis was moved only in the FIB view, while the beam shift in the X-Y axis was only used in the SEM view. Once the SEM and FIB views were well aligned, we chose a 40 × 40 μm ROI near the center of HVC while avoiding large synapse-free structures, such as blood vessels and cell

bodies. We then exposed the cross-sectional surface by milling a coarse trench into the tissue with a 16-nA ion beam. A fine polish was then performed with a 600 pA ion current. We fine-tuned the electron beam and imaged the exposed cross-section using the SmartSEM software and the EsB grid voltage set to 1.3 kV. We applied an angle correction to the acquired images to compensate for the tilting of the sample surface. We set the pixel size to 5 nm and the electron dwell time to 13 µs for all acquired FIBSEM imagery. 10-nm-thick layers were then serially milled away with a 600 pA Ga ion current, and the exposed block surfaces were serially imaged with a $8 \times 8$ µm square-shaped ROI window defined in the ATLAS 4D software (Fibics Incorporated). The auto-focus and auto-stigmator were used every 90 min to counter the milling-induced focus drift along the Z-axis.

## Synapse segmentation

We estimated the sizes of HVC synapses in FIBSEM imagery using the following segmentation procedure. The FIBSEM imagery was first aligned using TrakEM2 (*Cardona et al., 2012*) and then exported in tiff format. Because of the 2D alignment, the exported images contained margin areas with zero pixel values. We converted the values of these margin pixels to 255 (from black to white) using a custom macro implemented in Fiji. We then exported a subset of 200 consecutive images from each dataset to serve as training set for subsequent synapse segmentation.

We performed synapse segmentation using Ilastik, an interactive machine-learning toolkit for 3D image segmentation (*Sommer et al., 2011*). The FIBSEM training set and the entire dataset were first imported into Ilastik as single 3D volumes. We performed a semi-automatic pixel classification procedure followed by an object classification procedure (*Kreshuk et al., 2011*). Ilastik uses a random forest classifier (*Breiman, 2001*) to classify pixels or objects into different classes based on manually labeled objects defined as ground truth. In each training set, we labelled 5 categories of objects: synapses, mitochondria, membrane compartments, vesicles, and areas containing none of these features ('remainder'), *Figure 4—figure supplement 2a–b*. Large black objects were observed to interfere with synapse segmentation, which is why in a pre-processing step we changed the appearance of the image margins from black to white and why we categorized myelin sheaths and extracellular space as 'remainder'.

The initial labeling was performed on 20 slices that were evenly distributed across the training set. After each iteration of the supervised pixel classifier, we visualized the resulting pixel prediction map (*Figure 4—figure supplement 2c*) and then we corrected the ground truth until there were no major mistakes left (i.e. no missing synapses).

The smoothed pixel prediction map was then subjected to object classification. The object classifier only examined voxels with predicted synapse probabilities above 50%. It also excluded small objects containing less than 1000 interconnected voxels, which were too small to be synapses. The quality of the segmentation was visually verified after object classification.

After training and verification, the classifier was applied to the larger dataset that contained all of the images from a given bird. After inspection of the obtained synapse segmentations, the segmentations were exported as a binary image stack in multipage tiff files. The image stack had the same dimensions as the original input stack, its pixel values indicated whether a pixel belonged to a synapse (value 1) or not (value 0).

The FIBSEM datasets were of size 2–4 GB each. During synapse segmentation, Ilastik created several intermediate datasets that were   times the size of the original FIBSEM stacks. Therefore, adequate RAM and disk space were required to perform this analysis. We performed all analysis on a 2.8 GHz 6-Core Intel Xeon work-station with 24 GB RAM running Windows 7.

We wrote several MATLAB scripts to visualize, process, and analyze the segmentation results. Binary synapse segmentations were colored in red and superimposed on the original FIBSEM stack, *Figure 4—figure supplement 2d*. The superimposed image stack was visualized in MATLAB with a custom 3D image viewer. Each segmented synapse was recognized as an object composed of connected pixels and assigned an identification number. False positives (not synapses) and partial segmentations (that overlapped with the margin area) were automatically discarded. Falsely split or merged synapses were automatically identified and corrected. False negatives (missing synapses) were counted to ensure that no more than 3% of synapses were missed by our segmentation procedure, otherwise synapse segmentation was performed anew.

The corrected and recolored (blue: symmetric, red: asymmetric, *Figure 3b*) synapses were then subjected to volumetric and geometric measurements. The size of a synapse was defined as its

physical volume containing the pre- and post-synaptic membranes, their associated synaptic densities, and the synaptic cleft. The precise physical size of a single FIBSEM voxel was estimated using the procedure described in the Section on Physical Section Deformation.

## Synapse sizes

For the synapse size and Feret diameter measurements, on average 320 synapses were fully segmented and pooled in each bird (range 202–406 synapses per bird, n = 12 birds). Synapses were classified into asymmetric and symmetric subtypes and their sizes and diameters were measured. Histograms of logarithmically transformed synapse sizes were fitted with Gaussians, *Figure 4—figure supplement 1*. To determine how well synapse sizes could be fit with log-normal distributions, we performed goodness-of-fit tests (*Merchán-Pérez et al., 2014*). In all 12 birds, asymmetric synapses were log-normally distributed (p < 0.002 in every animal, Kolmogorov-Smirnov test). In 6/12 animals, symmetric synapses were log-normally distributed (p < 0.05, Kolmogorov-Smirnov test), whereas in the remaining six there was a trend for log-normal distribution (p < 0.3 in all six cases). Based on these findings, we decided to examine synapse sizes using log-transformed data.

For a given synapse $i$ with measured physical volume $V_i$, the log-transformed size $W_i$ is given by: $W_i = \log(V_i)$. Let $\mu = \frac{1}{N} \sum_i^N W_i$ be the mean of $W_i$ and $\sigma^2$ its variance, where $N$ is the total number of segmented synapses. The average synapse size $\bar{V}$ in a given bird is then given by $\bar{V} = e^\mu$. The standard error $SE$ of logarithmic synapse size is given by $SE = \frac{\sqrt{var}}{\sqrt{N}}$, where $var = \exp(2\mu + \sigma^2)(\exp(\sigma^2) - 1)$, see (*Mood et al., 1973*). Both $\bar{V}$ and $SE$ are displayed for each animal in *Figure 2B, C*.

We calculated the Feret diameters of synapses, defined as the diameter of the minimum-bounding sphere for each synapse. To improve the efficiency of this calculation, we implemented a linear time-randomized algorithm (*Welzl, 1991*). Average Feret diameters of both excitatory and inhibitory HVC synapses types were around 200 nm, which is much larger than the ssSEM-disector distance of 70 nm (we included an intermediate section in between the reference and look-up sections, the latter were separated by 140 nm, *Figure 6B*). Symmetric synapses had thinner synaptic clefts in the range 20 to 40 nm. Note that flat-shaped synapses parallel to the imaging plane with synaptic cleft size smaller than one third of the disector distance (70 nm / 3 ≈ 23.3 nm) could be missed by our disector counting. Therefore, it is possible that the disector counting systematically missed more symmetric synapses than asymmetric synapses. However, given the observed distribution of synapse sizes, we estimate to have missed at most 2.2 ± 0.7% of symmetric synapses (assuming random synapse orientations).

We obtained similar findings for synapse Feret diameters that were log-normally distributed in each animal for asymmetric synapses (p < 0.002, Kolmogorov-Smirnov test, n = 12 birds) and that were log-normally distributed in 7/12 animals for symmetric synapses (p < 0.05), with a trend for lognormal distribution in the remaining five animals (p < 0.31 in all five cases). Based on these findings, we performed across-group analysis of synapse sizes and synapse Feret diameters using log-transformed data that we fitted with linear mixed effect models.

## Linear mixed-effect models

We fitted linear mixed-effect (LME) models (*Gelman and Hill, 2007*) to our synapse data. We interpreted each measurement as a linear mixture of the following terms: a baseline value (usually representative of the ISO group), a fixed effect of tutoring or age, a random individual effect of the bird, and a zero-mean Gaussian observational error of fixed variance. We then evaluated the probability p (p value) that the fixed effect of tutoring or age significantly deviated from zero.

More specifically, for bird $m$ in the ISO group, the synapse density $y_{im}^{ISO}$ in disector $i$ was modeled as the sum of the fixed group mean $\beta^{ISO}$, the random individual-bird effect $b_m^{ISO}$, and the observation error $\epsilon_i^{ISO}$:

$$y_{im}^{ISO} = \beta^{ISO} + b_m^{ISO} + \epsilon_i^{ISO}.$$

The synapse densities $y_{im}^{SHORT}$ of birds in the SHORT group were modeled as

$$y_{im}^{SHORT} = \beta^{ISO} + \beta^{SHORT} + b_m^{SHORT} + \epsilon_i^{SHORT},$$

and the synapse densities $y_{im}^{LONG}$ of birds in the LONG group as

$$y_{im}^{LONG} = \beta^{ISO} + \beta^{LONG} + b_m^{LONG} + \epsilon_i^{LONG},$$

where $\beta^{SHORT}$ is the fixed effect of short-term tutoring on synapse densities and $\beta^{LONG}$ the fixed effect of long-term tutoring. In the Result Section, we report the fixed effect $\beta^{SHORT}$ and $\beta^{LONG}$ and their standard errors as obtained with the function *fitlmematrix* in Matlab (Mathworks Inc.). Tutoring had a significant effect on SHORT densities when the 95% confidence interval for $\beta^{SHORT}$ did not contain the solution $\beta^{SHORT} = 0$ (same for $\beta^{LONG}$). Reported p values correspond to the confidence of $\beta^{SHORT} \neq 0$ (same for $\beta^{LONG}$). Linear mixed-effect modeling of data from the second experiment was performed analogously.

## Acknowledgments

We thank John Anderson, Bopp Rita and German Koestinger for assistance and advice on neuro-anatomy and electron microscopy data, Homare Yamahachi and Daniel Düring for their help with the manuscript, Cassandra Pattanayak for statistical advice, and Anna Kreshuk and Stuart Berg of the Ilastik team in Heidelberg for advice on using Ilastik for synapse segmentation. We also would like to thank Todd Roberts and Yoko Yazaki-Sugiyama for their helpful discussions about the science.

## Additional information

### Funding

| Funder | Grant reference number | Author |
|---|---|---|
| Susan Todd Horton Class of 1910 Trust | | Houda G Khaled |
| Hubel Neuroscience Summer Research Fellowship | | Houda G Khaled |
| Seven College Conference Junior Year Abroad Award | | Houda G Khaled |
| Wellesley College | Faculty Awards F11, F12, S14, F14, S16 | Sharon MH Gobes |
| National Institutes of Health | R15HD085143 | Sharon MH Gobes |
| Schweizerischer Nationalfonds zur Förderung der Wissenschaftlichen Forschung | 31003A_127024 | Richard HR Hahnloser |
| Schweizerischer Nationalfonds zur Förderung der Wissenschaftlichen Forschung | 31003A_156976 | Richard HR Hahnloser |
| Schweizerischer Nationalfonds zur Förderung der Wissenschaftlichen Forschung | ZKOZ3_160663 | Richard HR Hahnloser |
| European Research Council | FP7/2007-2013 / ERC Grant AdG 268911 | Richard HR Hahnloser |
| ETH Zürich Foundation | Project 2015-48 3 | Richard HR Hahnloser |
| Schweizerischer Nationalfonds zur Förderung der Wissenschaftlichen Forschung | IZKOZ3_160663 | Richard HR Hahnloser |

The funders had no role in study design, data collection and interpretation, or the decision to submit the work for publication.

## Author contributions
Ziqiang Huang, Conceptualization, Data curation, Software, Formal analysis, Investigation, Methodology, Writing—original draft, Writing—review and editing; Houda G Khaled, Data curation, Formal analysis, Investigation; Moritz Kirschmann, Conceptualization, Resources, Methodology; Sharon MH Gobes, Conceptualization, Formal analysis, Supervision, Methodology, Writing—review and editing; Richard HR Hahnloser, Conceptualization, Resources, Data curation, Software, Formal analysis, Supervision, Funding acquisition, Validation, Investigation, Visualization, Methodology, Writing—original draft, Project administration, Writing—review and editing

## Author ORCIDs
Ziqiang Huang http://orcid.org/0000-0002-4185-0947
Houda G Khaled http://orcid.org/0000-0002-0759-0272
Sharon MH Gobes http://orcid.org/0000-0001-9598-7116
Richard HR Hahnloser http://orcid.org/0000-0002-4039-7773

## Ethics
Animal experimentation: All experimental procedures were in accordance with the Veterinary Office of the Canton of Zurich (207-2013).

## Decision letter and Author response
Decision letter https://doi.org/10.7554/eLife.37571.022
Author response https://doi.org/10.7554/eLife.37571.023

## Additional files

### Supplementary files
• Transparent reporting form
DOI: https://doi.org/10.7554/eLife.37571.018

### Data availability
We provide all SSEM synaptic density data for Experiments I and II in the Matlab file ssSEM_exp1and2_groupSeperated.mat. We provide all FIBSEM data for Experiment I in the Matlab file FIBSEM_exp1.mat. HVC volume data for Experiment II is provided in the Matlab file HVCvolume_exp2.mat. To reproduce our linear mixed effects analyses, we provide the Matlab function getLME. For example, to reproduce the comparison between synaptic densities in LONG and LONG60 birds, one first needs to load the data: load ssSEM_exp1and2_groupSeperated, then one needs to concatenate the relevant variables: data=vertcat(data_ssSEM_exp1_LONG,data_ssSEM_exp2_TUT60), and finally, one needs to run the function: getLME(data), followed by typing 1 for running the analysis for asymmetric synapses for example. All Matlab files can be retrieved from https://www.research-collection.ethz.ch/handle/20.500.11850/285394, DOI 10.3929/ethz-b-000285394.

The following dataset was generated:

| Author(s) | Year | Dataset title | Dataset URL | Database and Identifier |
| --- | --- | --- | --- | --- |
| Huang Z, Hahnloser R | 2018 | Synapse density in HVC of Zebra Finches: Data and Matlab Code | https://www.research-collection.ethz.ch/handle/20.500.11850/285394 | ETH Zürich Research Collection, 10.3929/ethz-b-000285394 |

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
