## [Decision Letter]

Thank you for submitting your article "Critical sensory experience rapidly imbalances excitatory and inhibitory synapses for vocal control" for consideration by *eLife*. Your article has been reviewed by two peer reviewers, and the evaluation has been overseen by a Reviewing Editor and a Senior Editor. The reviewers have opted to remain anonymous.

The reviewers have discussed the reviews with one another and the Reviewing Editor has drafted this decision to help you prepare a revised submission.

Summary:

This manuscript uses serial section electron microscopy to count excitatory (asymmetric) and inhibitory (symmetric) synapses in the zebra finch HVC. They measure synapse numbers in birds isolated from song learning, short song learning and normal exposure, then perfuse the birds at the end of the critical period. The main finding is that song learning is associated with changes in the balance of excitatory and inhibitory synapses.

Essential revisions:

The reviewers were divided on your paper and have discussed it extensively. We all agreed that the results of the study are clear and acceptable, and we appreciate quantification of the finding that the balance of excitatory and inhibitory synapses changes in HVC during the course of song learning.

Nevertheless, there is a serious concern about the use of the dissector counting method without enough objects present in many probes. You claim to resolve this difficulty, but it was not clear to us how this was done (see below). There is a very rich literature on using stereological methods, and it's clear that, if no objects are counted, then the probes used are not large enough. The divide by zero problem should not exist, if done properly. It was also not clear why a dissector method was used when some of the analysis was done with volume EM imaging. Volume imaging should allow for estimates of synapse size, which would be important verification that a dissector method could even be used. If there is only a change in synapse size between conditions and not numbers, then the dissector method will give a false result and show density change.

We found your jack-knife estimate method to be unconvincing, and would suggest that if dissectors are chosen that provide no counts they are of incorrect dimensions. If you would like to stick to this approach, please provide more support. For example, if you use ssSEM and FIBSEM, could you not verify your estimating method with another counting method?

There were other significant concerns about the interpretation of the data. Your counts provide unique and potentially valuable measure of synapse density in HVC, but cannot be used to show causality. Thus, the statement that "exposure to tutor song leads to elimination of excitatory synapses in HVC" implies causality, where none is shown. There are many examples of this.

Your hypothesis, that there is HVC synapse turnover associated with aging, is possible, but you do not address turnover. The stated idea, that extended tutoring had an effect on synapse pruning, can only be hypothesized. Turnover may be inferred but is not shown here.

There are also a number of unnecessary overstatements, with an example from the Abstract "little is known about their formation during critical periods" when there is a large literature on the role of inhibition in visual system critical periods. There is no need to address this specific example in the reply to the reviews. Instead, if the authors choose to revise the paper, they should carefully go through the paper and remove all overstatements, since they detract from your argument.

---

## [Author Response]

Essential revisions:The reviewers were divided on your paper and have discussed it extensively. We all agreed that the results of the study are clear and acceptable, and we appreciate quantification of the finding that the balance of excitatory and inhibitory synapses changes in HVC during the course of song learning.Nevertheless, there is a serious concern about the use of the dissector counting method without enough objects present in many probes. You claim to resolve this difficulty, but it was not clear to us how this was done (see below). There is a very rich literature on using stereological methods, and it's clear that, if no objects are counted, then the probes used are not large enough. The divide by zero problem should not exist, if done properly.

The divide-by-zero problem stems from the small size of dissectors. However, none of this is a real problem, because dissectors can be combined to larger units in which fractions of objects can be estimated. The dissector size is like the sample time in electrophysiology. The sample time is usually very small such that in each sample bin there is at most a single spike. Nevertheless, it is possible to estimate the fraction of spikes in a neuron that occur in bursts. The situation is exactly the same with dissectors, they are simply the sample unit. In the same way that the fraction of spike bursts can be reliably estimated regardless of the sampling rate, the fraction of inhibitory synapses can be reliably estimated regardless of dissector size. The trick is to consider many same sample bins, i.e., many dissectors.

We combined dissectors using the jackknife (leave-one-out) estimate of variance, which is known to be highly conservative. We thus used a standard method for variance estimation, which leaves no room for mistakes. We are thus confident that our findings do not depend on our particular choice of dissectors. See also below for an independent estimate of variance.

On a more technical level, the dissector method is very robust to dissector size thanks to distinction between inclusion and exclusion boundaries. We have included a new Figure (Figure 6) in the manuscript to better explain the dissector method.

In experiment I, we inspected on average 96 dissectors per animal (range 68-111 dissectors, n=12 birds). We counted on average 188 dissected synapses per bird (range 120-245, n=12 birds), among which on average 42 were symmetric (range 26-68, n=12 birds). In experiment II, we inspected on average 102 dissectors per animal (range 77-124 dissectors, n=20 birds). We counted on average 175 synapses per bird (range 114-250, n=20 birds), among which on average 29 were symmetric (range 5-49, n=20 birds). We have added these numbers to the Materials and methods section of our manuscript.

It was also not clear why a dissector method was used when some of the analysis was done with volume EM imaging. Volume imaging should allow for estimates of synapse size, which would be important verification that a dissector method could even be used. If there is only a change in synapse size between conditions and not numbers, then the dissector method will give a false result and show density change.

A strength of the dissector method is robustness to the shape of objects to be counted, including the objects’ size. The only problem that can arise with the dissector method is when objects are too small, such that many objects fall in between the reference and look-up sections (see new Figure 6C). Actually we had considered this possibility and included an intermediate section in between the reference and the look-up sections to inspect such cases (Figure 6A). We found that the average Feret diameters of both excitatory and inhibitory HVC synapses types were around 200 nm, about 3 times as large as distance of 70 nm. In our manuscript, we wrote: ‘Dissectors provide reliable estimates of object numbers (errors smaller than 6%) when the separating distance between consecutive dissectors is no larger than twice the mean size of objects. When the separating distance is larger than three times the mean object size, density errors rapidly increase to 27% (Merchán-Pérezet al., 2009)‘. Thus, given these numbers, synapses are huge compared to the dissector distance and our use of volume EM imaging confirms the validity of using the dissector method.

As detailed in the Materials and methods section, we estimated the dissector method to systematically miss more symmetric synapses than asymmetric synapses, but in only 2.2 ± 0.7% of cases (given the observed distribution of synapse sizes and their random orientation). Furthermore, as we had shown, the size of symmetric synapses does not vary among the various bird groups. Thus, there is no reason to believe that the dissector method was used inappropriately.

We found your jack-knife estimate method to be unconvincing, and would suggest that if dissectors are chosen that provide no counts they are of incorrect dimensions. If you would like to stick to this approach, please provide more support. For example, if you use ssSEM and FIBSEM, could you not verify your estimating method with another counting method?

To convince ourselves that our findings do not depend on the particular choice of how to combine dissectors (i.e. using jackknife), we also analyzed synapse ratios using a linear mixed-effect (LME) model. Namely, in each animal we combined all dissectors into four distinct sets, providing us with four independent estimates of percent inhibitory synapses. We then fit a LME model to these estimates, with a common fixed common effect (offset) for each dissector set, a fixed effect for short tutoring, a random individual effect for each bird, plus a Gaussian observation error. When applying this approach to ISO and SHORT birds, we find an offset of βiso=0.19 and a nonzero fixed effect of tutoring βshort=0.08 that was highly significant ((βshort≠0, p=1.7*10^-6^, 32 observations, 2 fixed effect coefficients, 8 random effect coefficients).

Note that nearly identical numbers had been obtained with jackknife estimation. The p-value we obtained with jackknife+bootstrapping was slightly larger (p=5*10^-5^). For the comparison between ISO and LONG birds, we obtained βlong=0.01, which was not significantly different from zero (p=0.48), again in agreement with our jackknife analysis in the manuscript.

We decided in the revised manuscript to report in the Results section the new LME method of variance and p-value estimation (to keep with the general strategy of using LME), and to report the jackknife analysis in the Materials and methods.

There were other significant concerns about the interpretation of the data. Your counts provide unique and potentially valuable measure of synapse density in HVC, but cannot be used to show causality. Thus, the statement that "exposure to tutor song leads to elimination of excitatory synapses in HVC" implies causality, where none is shown. There are many examples of this.

We are struggling to understand why our work should not show causality. In fact, we are convinced that it shows causality. The only difference between ISO and SHORT birds is one day of tutor exposure. So what else than tutor exposure can have caused the changes in synaptic density that we observed? We believe we kept all other variables constant between the groups.

Our language of implied causality seems to agree with typical usage in the literature, check for example the paper title of (Knott et al., 2002): ‘Formation of Dendritic Spines with GABAergic Synapses Induced by Whisker Stimulation in Adult Mice’, which clearly entails causality (‘induced’).

Of course, causality does not imply that the mechanism for the observed effect is known and maybe the reviewer means mechanism, not causality. Biologists strive to identify mechanisms, but they are well aware that hardly any (causal) finding published in the literature is understood at the full mechanistic level (not even considering quantum mechanics). Causality and mechanism are two different things, it makes sense only to investigate mechanisms of effects for which causality has been shown.

We have toned down the implied causality at several places including the title. We mean to say that we have no problem with changing our wording everywhere to remove implied causality, but we feel we must be given a reason by the reviewers why we should do so – we would then happily discuss the shortcomings of our approach, as these clearly must point to very important future work that should be done to show causality.

Your hypothesis, that there is HVC synapse turnover associated with aging, is possible, but you do not address turnover. The stated idea, that extended tutoring had an effect on synapse pruning, can only be hypothesized. Turnover may be inferred but is not shown here.

We agree that we do not have direct evidence of turnover. However, turnover is suggested from previous work including (Roberts et al., 2010). What matters in our study is the net effect of overall pruning or insertion of synapses, irrespective of whether it is accompanied by turnover. We have rephrased all sentences mentioning turnover.

There are also a number of unnecessary overstatements, with an example from the Abstract "little is known about their formation during critical periods" when there is a large literature on the role of inhibition in visual system critical periods. There is no need to address this specific example in the reply to the reviews. Instead, if the authors choose to revise the paper, they should carefully go through the paper and remove all overstatements, since they detract from your argument.

We have gone through the paper and hopefully eliminated all such overstatements. Note that our sentence in the Abstract dealt with the development of motor skills, for which visual cortex plays no particular role.